# Functional Characterization of Allelic Variations of Human Cytochrome P450 2C8 (V181I, I244V, I331T, and L361F)

**DOI:** 10.3390/ijms24098032

**Published:** 2023-04-28

**Authors:** Yoo-Bin Lee, Vitchan Kim, Sung-Gyu Lee, Gyu-Hyeong Lee, Changmin Kim, Eunseo Jeong, Donghak Kim

**Affiliations:** Department of Biological Sciences, Konkuk University, Seoul 05025, Republic of Korea

**Keywords:** cytochrome P450, P450 2C8, allelic variant, paclitaxel, arachidonic acid, mass spectrometry

## Abstract

The human cytochrome P450 2C8 is responsible for the metabolism of various clinical drugs as well as endogenous fatty acids. Allelic variations can significantly influence the metabolic outcomes. In this study, we characterize the functional effects of four nonsynonymous single nucleotide polymorphisms *15, *16, *17, and *18 alleles recently identified in cytochrome P450 2C8. The recombinant allelic variant enzymes V181I, I244V, I331T, and L361F were successfully expressed in *Escherichia coli* and purified. The steady-state kinetic analysis of paclitaxel 6-hydroxylation revealed a significant reduction in the catalytic activities of the V181I, I244V, and L361F variants. The calculated catalytic efficiency (*k*_cat_/*K*_m_) of these variants was 5–26% of that of the wild-type enzyme. The reduced activities were due to both decreased *k*_cat_ values and increased *K*_m_ values of the variants. The epoxidation of arachidonic acid by the variants was analyzed. The L361F variant only exhibited 4–6% of the wild-type catalytic efficiency in ω-9- and ω-6-epoxidation reactions to produce 11,12-epoxyeicosatrienoic acid (EET) and 14,15-EET, respectively. These reductions were mainly due to a decrease in the *k*_cat_ value of the L361F variant. The binding titration analysis of paclitaxel and arachidonic acid showed that all variants had similar affinities to those of the wild-type (10–14 μM for paclitaxel and 20–49 μM for arachidonic acid). The constructed paclitaxel docking model of the variant enzyme suggests that the L361F substitution leads to the incorrect orientation of paclitaxel in the active site, with the 6′C of paclitaxel displaced from the productive catalytic location. This study suggests that individuals carrying the newly identified P450 2C8 allelic variations are likely to have an altered metabolism of clinical medicines and production of fatty acid-derived signal molecules.

## 1. Introduction

Cytochrome P450 (P450, CYP) enzymes are a superfamily of heme-thiolate proteins that catalyze the oxidative metabolism of a wide range of exogenous chemicals and endogenous substances [1]. Many human P450 enzymes metabolize different substrates, and their catalytic activities are subject to a variety of regulatory influences, such as genetic polymorphisms, and hormonal and environmental factors [2].

The P450 2C subfamily comprises approximately 20% of the total P450 content in liver microsomes and metabolizes approximately 20% of clinically used drugs on the market [3,4]. Humans have four P450 members, 2C8, 2C9, 2C18, and 2C19, which are located on chromosome 10 (10q24.1–q24.3) [5]. Among these, P450 2C8 is expressed at a relatively high level in the liver, comprising approximately 7% of the total hepatic P450 proteins [6,7]. P450 2C8 is involved in the metabolism of important therapeutic drugs including the anticancer drug paclitaxel. P450 2C8 is also expressed in the kidney and vasculature, where it converts the endogenous fatty acid arachidonic acid (AA) to generate biologically active epoxyeicosatrienoic acids (EETs). EETs are involved in the regulation of blood pressure, hepatic glycogenolysis, the secretion of peptide hormones in the pancreas and pituitary, and the inhibition of vascular smooth muscle cell migration in the kidneys [8,9,10,11,12].

Eighteen allelic variants of P450 2C8 have been identified to date (https://www.pharmvar.org/gene/P450 2C8, accessed on 28 March 2023). Recently, Gaedigik et al. reported four novel alleles of P450 2C8, for which the functional consequences are unknown [13]. The aim of this study is to characterize the in vitro functional activities of four P450 2C8 nonsynonymous nucleotide polymorphisms (V181I, I331T, I244V and L361F), which occur in P450 2C8*15, *16, *17, and *18 alleles. 

## 2. Results

### 2.1. Expression and Purification of the Recombinant P450 2C8 Variant Enzymes

The expression vectors for the four P450 2C8 variant enzymes were constructed in pCW bicistronic expression vectors, and the recombinant proteins were successfully expressed in *E. coli*, and all of the mutant enzymes displayed typical CO-binding Soret spectra at 450 nm, with no peak at 420 nm (Figure 1A). The expression levels of P450 2C8 *15, *16, *17, and *18 ranged from 428 to 684 nmol/L, which was similar to or slightly lower than that of the wild-type (730 nmol/L culture) (Figure 1A). The purification of the recombinant P450 2C8 variant enzymes was successfully performed using Ni^2+^-NTA column chromatography (Figure 1B). 

### 2.2. Substrate Binding Affinities

The binding affinities of the purified P450 2C8 variants to paclitaxel and arachidonic acid were determined (Figure 2 and Table 1). The titrations of both substrates displayed typical type-I spectral changes (Figure 2). All variant enzymes, except I331T, demonstrated similar binding affinities to paclitaxel as the wild-type enzyme, while an increased *K*_d_ value was observed for I331T, indicating a small decrease in the substrate binding affinity (Table 1). This result indicated that the binding affinities of mutants to paclitaxel did not affect their catalytic activities. In the binding analysis of arachidonic acid, two variants (I331T and I244V) exhibited a significant decrease in the binding affinities with 2–2.5-fold increased *K*_d_ values (Table 1). This result suggests that the altered binding affinities of the I331T and I244V mutants may influence their catalysis of arachidonic acid epoxidation. 

### 2.3. Catalytic Activities of the P450 2C8 Allelic Variants 

P450 2C8 (wild-type) catalyzes the oxidative metabolism of paclitaxel to produce 6α-hydroxy paclitaxel as the main metabolite. The chromatogram of the UPLC-tandem mass analysis revealed 6α-hydroxy paclitaxel as the main metabolite at a retention time of 3.6 min (Figure 3A). The steady-state kinetic analysis of the P450 2C8 wild-type indicated *a k*_cat_ value of 0.81 min^−1^ and *a K*_m_ value of 4.9 μM (Figure 4 and Table 2). Three P450 2C8 variants (V181I, I331T, and L361F) displayed reduced catalytic efficiencies for paclitaxel (Figure 4 and Table 2). In particular, the L361F variant exhibited a significant decrease in *k*_cat_ and an increase in *K*_m,_ resulting in a catalytic efficiency of only 5% of that of the wild-type enzyme. However, the I244V variant exhibited a significantly enhanced catalytic efficiency, with an increased *k*_cat_ value (Figure 4 and Table 2). This result suggests that the mutations in three variants (V181I, I331T, and L361F) may produce the altered clinical outcomes in the drug metabolism of P450 2C8. 

As an endogenous substrate of P450 2C8, the epoxidation of arachidonic acid was analyzed. The wild-type enzyme catalyzes ω-9 and ω-6 epoxygenase reactions to produce two main metabolic products, 11,12-epoxyeicosatrienoic acid (EET) and 14,15-EET (Figure 3B). The kinetic parameters of the wild-type enzyme were obtained, with a *k*_cat_ of 14.6 min^−1^ and a *K*_m_ of 8.8 μM for ω-9 epoxidation, and a *k*_cat_ of 35.1 min^−1^ and a *K*_m_ of 13.3 μM for ω-9 epoxidation (Table 3). All variants, except I244V, displayed reduced catalytic efficiencies, mainly due to decreased *k*_cat_ values for both ω-9 and ω-6 epoxidation reactions (Figure 5 and Table 3). In particular, the L361F variant also displayed a dramatic reduction in reaction activity, with a 23-fold decrease for ω-9 epoxidation and an 18-fold decrease for ω-6 epoxidation (Table 3). Interestingly, the I244V variant exhibited a similar or slightly higher catalytic activity for arachidonic acid. This result suggests that the substitution of Ile at position 244 with the non-polar side chain Val did not alter the catalytic function of P450 2C8. This result suggests that the individuals possessing the genetic variation of P450 2C8, particularly L361F, may be deficient in the production of endogenous EETs.

## 3. Discussion

Novel allelic variant enzymes of P450 2C8 were expressed and analyzed to study the enzymatic changes resulting from altered residues that affect the clinical and endogenous substrates paclitaxel and arachidonic acid. These alleles were identified using next-generation sequencing (NGS) and haplotype analyses targeting P450 2C8, 2C9, and 2C19 [13]. In this study, each allele was found to exhibit altered enzyme activity ranging from reduced to increased function, leading to a wide range of phenotypic ranges between individuals and populations. In addition, the substrate preference of these variant enzymes differed from that of the wild-type enzyme. Thus, some clinically important changes in the P450 metabolism can result from genetic polymorphisms of P450 2C8. Previous clinical studies have reported considerable inter-individual variations in the response to paclitaxel in cancer patients [14,15]. To date, 18 allelic variants of P450 2C8 have been identified, and among them, the enzymatic functions of the 4 variants in this study were not identified. P450 2C8 *15 (V181I) and P450 2C8 *18 (L361F) were found in Caucasians, P450 2C8 *16 (I331T) was found in African Americans, and P450 2C8 *17 (I244V) was found in Yoruba, respectively [13]. 

The crystal structures of P450 2C8 complexed with various substrates, including montelukast, felodipine, troglitazone, and 9-cis-retinoic acid, revealed a large active site cavity of the P450 enzyme [6,16]. These structural analyses indicate that P450 2C8 possesses a versatile substrate-binding site architecture that is adaptable to many drugs as well as endogenous substrates with different chemical structures. The L361F mutation is located in the loop between helices J and K of P450 2C8. To analyze the effects of the L361F variation, molecular docking models with paclitaxel were constructed using the X-ray crystal structure of P450 2C8 (PDB ID 1PQ2) (Figure 6). Among several docking models, the optimal model was chosen based on the substrate heading to the heme group for a productive response with the minimum energy and lowest RMSD. The active site of P450 2C8 in the wild-type enzyme illustrated that the C6 position of paclitaxel was projected to the heme at a distance of 5.8 Å, indicating a productive substrate orientation (Figure 6A). However, the L361F docking model with the lowest binding energy indicated that the Phe361 residue, which has a large size, occupied the space above the heme to move the C6 position away from the active sites (Figure 6B). This docking model suggested that the L361F variant positioned the substrate in a non-productive orientation to result in a dramatic decrease in enzyme catalytic activity, while its substrate binding was still maintained, indicating a similar affinity to the wild-type (Table 1). In a previous study, the Halpert group reported the role of the L362F mutation of P450 2B1 in the oxidation of steroids and 7-alkoxylcoumarins [17]. The amino acid sequence alignment indicated that the L362F mutation of P450 2B1 corresponded to the L361F mutation of P450 2C8 (Appendix A). The L362F mutant enzyme of P450 2B1 exhibited significant decreased catalytic rates in the oxidation reactions of testosterone and 7-butoxycoumarin [17]. This result suggests that the substitution of Leu362 residue with Phe in the P450 2 subfamily enzymes could result in dramatic alteration of catalytic activity. The I244V mutation was located in helix G of P450 2C8. The structures of the F–G region form the outside boundary of the substrate-binding cavity and channel architecture of P450 enzymes, which differs between mammalian P450s in accordance with the flexibility of P450. In the structure of P450 2C8, the plasticity of the helix F–G region is comparatively low, as it forms a dimer with other P450 2C8 molecules nearby helices [6]. Since Ile and Val are both hydrophobic amino acids with similar molecular weights, the substitution of Ile with Val in the F–G helices may not alter the substrate binding or catalytic activity of the variant much (Table 1, Table 2 and Table 3). The constructed docking model of the I244V mutation indicated that there was no possibility of a dramatic conformational change in the stereoscopic structure by substitution (Appendix A). The mutation of V181I was located at the C-terminus of helix E of the P450 2C8 structure (Appendix A) [6]. The Val181 residues in helix E appear to play a role in interacting with helix G and the N-terminus of helix H [6]. Therefore, the additional methyl chain of the V181I mutant could produce the extra space between helices and perturbate the optimal orientation of helices E, G, and H, despite the substitution with a residue that is also hydrophobic. Meanwhile, the I331T mutation was found at the C-terminus of the J-helix [6]. The Ile331 reside is close to the Arg333, which may be one of the basic residues interacting with the NADPH-P450 reductase (Appendix A) [18]. One postulation of the I331T mutation is that it could affect the reductase binding and electron transfer step, which should be further analyzed. 

In the steady-state kinetic analysis of arachidonic acid epoxidation, the V181I mutant displayed relatively high errors in both *K*_m_ values, and therefore, the parameters of its catalytic efficiencies (*k*_cat_/*K*_m_) were also influenced. We performed the repeated experiments with more data points but the enzyme activities of the V181I mutant was decreased in more than 30 μM of substrate concentration. The inhibitory effect of high substrate concentration on the V181I mutant needs to be addressed in further studies. 

In previous studies, ten nonsynonymous single-nucleotide polymorphisms (SNPs) in the coding region of CYP2C8 were reported [19]. Interestingly, P450 2C8*2 (I269F) was found only in African Americans, at a frequency of 0.18%, while P450 2C8*3 (R139K; K399R) and *4 (I264M) occurred primarily in Caucasians, at frequencies of 0.075% and 0.13%, respectively [12,20]. They display significantly reduced catalytic activities toward paclitaxel and arachidonic acid [12]. Seven P450 2C8 alleles, namely, *6 (G171S), *7(R186X), *8 (R186G), *9 (K247R), *10 (K383N), *13 (K383N), and *14 (A238P), were found in Japanese populations [21,22]. Alleles *8 (R186G) and *14 (A238P) displayed decreased activity, and allele *7 (R186X) exhibited no enzyme activity [21,22]. Interestingly, the *9 (K247R) allele did not alter the activity [21]. This mutated residue is close to that found at residue Ile244 in the *16 allele; therefore, the I244V mutation in this study may not affect catalytic activity.

The purified enzymes displayed little shoulder peaks at 420 nm (Figure 1B) and the similar peak at 420 nm was observed in the purified wild-type CYP2C8 enzyme. These small peaks of 420 nm in the purified enzymes (WT and mutants) seemed to be originated from the purification procedures, and the structural stability of the P450 protein was not affected in any of the variants.

## 4. Materials and Methods

### 4.1. Chemicals

Arachidonic acid, 3-[(3-cholamidopropyl)dimethylammonio]-1-propanesulfonate (CHAPS), 1,2-dilauroyl-*sn*-glycero-3-phosphocholine (DLPC), 5-aminolevulinic acid (ALA), NADP^+^, and NADPH were purchased from Sigma–Aldrich (St. Louis, MO, USA). Paclitaxel was purchased from Santa Cruz Biotechnology (Dallas, TX, USA), and the primary vascular eicosanoid LC–MS mixture and (±)11,12-epoxy-5Z,8Z,14Z-eicosatrienoic acid (11,12-EET) were purchased from Cayman Chemicals (Ann Arbor, MI, USA). 6α-Hydroxypaclitaxel was kindly provided by Professor Sang Kyum Kim of Chungnam National University. Ni^2+^-nitrilotriacetate (NTA) agarose was purchased from Qiagen (Hilden, Germany). Other chemicals of the highest commercially available grade were used. *Escherichia coli* DH5α was purchased from Invitrogen (Carlsbad, CA, USA). Rat NADPH-P450 reductase (NPR) was heterologously expressed in *E. coli* (TOPP3 strain) and purified as described previously [23]. 

### 4.2. Construction of P450 2C8 Variants

P450 2C8 wild-type cDNA cloned into pCW bicistronic plasmid was used as a template to generate point mutations at target sites. The pCW/P450 2C8 plasmid previously described for the expression of the membrane-bound enzyme in *E. coli* was modified with a C-terminal hexahistidine tag to express the enzyme [24]. The ligated pCW/P450 2C8-His gene was verified using nucleotide sequencing. 

To construct the four P450 2C8 mutants, site-directed mutagenesis was performed using a QuikChange^®^ site-directed mutagenesis kit (Agilent Technologies, Santa Clara, CA, USA). The primers used for site-directed mutagenesis were P450 2C8*15 (V181I; 5′-GC AAT GTG ATC TGC TCC ATT GTT TTC CAG AAA CG-3′), P450 2C8*16 (I331T; 5′-GAA GAG ATT GAT CAT GTA ACC GGC AGA CAC AGG-3′), P450 2C8*17 (I244V; 5′-GCT CTT ACA CGA AGT TAC GTT AGG GAG AAA G-3′), and P450 2C8*18 (L361F; 5′-CAG AGA TAC AGT GAC TTT GTC CCC ACC GGT GTG CC-3′). The entire coding region, including the mutated sites, was isolated and verified using nucleotide sequencing to confirm the successful incorporation of the desired mutagenesis without extraneous changes. The cDNAs of the V181I, I331T, I244V, and L361F variants were subcloned into a new pCW bicistronic expression vector for expression. 

### 4.3. Expression and Purification of P450 2C8 Variant Enzymes

Plasmids carrying the cDNA of the P450 2C8 genetic variants were transformed into *E. coli* DH5α competent cells. Single isolated colonies were grown with shaking at 37 °C overnight in LB medium containing 50 µg/mL ampicillin. After overnight incubation, each clone was transferred into 500 mL of Terrific Broth containing 50 µg/mL ampicillin and incubated at 37 °C using a 230 rpm shaker until the OD600 of the culture reached 0.4–0.6. The supplements 0.5 mM 5-ALA, 1.0 mM thiamine, 1.0 mM IPTG, and trace elements were added to induce CYP enzyme expression. The cultures were then grown at 28 °C with shaking at 200 rpm for 24 h and the cells were harvested using centrifugation at 4 °C for 30 min. The P450 2C8 genetic variants containing a C-terminal hexahistidine-tag were prepared in the bicistronic membrane fraction, and expression of the P450 2C8 enzymes in *E. coli* DH5α was accomplished as described previously [25,26]. The harvested cells were resuspended in TES buffer containing lysozyme and incubated for 30 min at 4 °C. The resuspended cells were centrifuged at 4000 rpm for 30 min and sonicated in 200 mL of sonication buffer containing dithiothreitol. After ultracentrifugation, the bacterial membrane fractions were prepared and then solubilized overnight in solubilization buffer containing 0.1 mM EDTA, 1.5% CHAPS (*w*/*v*), 10 mM β-mercaptoethanol, and 20% glycerol at 4 °C. The solubilized materials were then purified using a Ni^2+^-nitrilotriacetate column according to a previously described method [27]. The recombinant P450 2C8 proteins were eluted from the column using a buffer containing imidazole 300 mM. The eluted fractions containing the P450 2C8 variants were dialyzed against 100 mM potassium phosphate buffer (pH 7.7) containing 20% glycerol (*v*/*v*), 1.0 mM EDTA, and 0.10 mM dithiothreitol at 4 °C for 24 h.

### 4.4. Substrate Binding Assay

Binding titration analysis was performed using the purified P450 2C8 wild-type and variant enzymes. The P450 2C8 genetic variants were diluted to 2 µM in 100 mM potassium phosphate buffer (pH 7.4) and divided into 2 glass cuvettes. Spectroscopic changes (350–500 nm) were measured by adding the substrate paclitaxel or arachidonic acid (dissolved in CH_3_CN or methanol, respectively) using a Cary 100 spectrophotometer (Varian, Inc., Palo Alto, CA, USA). The difference between the maximum (385 nm) and minimum (425 nm) absorbance was plotted against the substrate concentration to calculate the substrate-binding affinity (*K*_d_).

### 4.5. Catalytic Activity Assays

The catalytic activities of the recombinant wild-type and mutant P450 2C8 enzymes were determined using a P450/NPR/phospholipid reconstituted system, and the P450 2C8 metabolism of paclitaxel was analyzed by monitoring 6α-OH paclitaxel formation using ultra-performance liquid chromatography–tandem mass spectrometry (UPLC–MS/MS) (Waters, Milford, MA, USA). The reconstitution mixtures consisted of 50 pmol of purified P450 2C8 genetic variants, 100 pmol of purified rat NADPH reductase, paclitaxel (0, 1, 5, 10, 50, and 100 µM), DLPC, and 100 mM potassium phosphate buffer (pH 7.4) in a final volume of 500 µL. All assays were performed in duplicate. After 3 min of pre-incubation at 37 °C for temperature equilibration, the reaction was initiated by addition of the NADPH-generating system. All reaction mixtures were incubated for 5 min at 37 °C and terminated by the addition of 1 mL CH_2_Cl_2_. The samples were vortexed and centrifuged at 3500 rpm for 15 min. The organic phase of each sample was extracted into a clean test tube, evaporated under a stream of N_2_ gas, and resuspended in 100 µL CH_3_CN for LC-MS/MS quantification. To analyze paclitaxel hydroxylation, the samples were injected into an ACQUITY UPLC™ BEH C18 column (50 × 2.1 mm, 1.7 µm) equipped with Waters Quattro Premier™ and Waters ACQUITY UPLC™. The mobile phase consisted of 0.1% formic acid in H_2_O:CH_3_CN (90:10 *v*/*v*) (A) and 0.1% formic acid in CH_3_CN (B). A gradient mobile phase B was applied at 10% during the first 0.5 min and then increased to 100% for 4 min. The flow rate was 0.3 mL/min. Mass spectra were acquired using positive electrospray ionization (ESI+) in multiple reaction mode (MRM). The source temperature was 120 °C, the desolvation temperature was 380 °C, the cone gas flow rate was 50 L/h, and the desolvation gas flow rate was 550 L/h. The column temperature was set at 40 °C. The positive ionization transitions of paclitaxel (*m*/*z* 854.4 > 286) and 6α-hydroxypaclitaxel (*m*/*z* 870.4 > 286) were monitored at collision energies of 20 and 24 eV, respectively. QuanLynx software (Waters) was used for peak area calculation and analysis.

The reconstitution mixtures consisted of 100 pmol of the purified P450 2C8 genetic variants, 200 pmol of purified rat NADPH reductase, arachidonic acid (0, 3, 5, 10, 20, 30, and 50 µM), DLPC, and 100 mM potassium phosphate buffer (pH 7.4) in a final volume of 500 µL. All assays were performed in duplicate. After 3 min pre-incubation at 37 °C for temperature equilibration, the reaction was initiated by adding the NADPH-generating system. All enzymatic reaction mixtures were incubated at 37 °C for 10 min and terminated by the addition of 1 mL of CH_2_Cl_2_. The samples were vortexed and centrifuged at 3500 rpm for 15 min. The organic phase of each sample was extracted into a clean test tube, evaporated under a stream of N_2_ gas, and resuspended in 100 µL CH_3_CN for LC-MS/MS quantification. To analyze arachidonic acid epoxidation, the samples were injected into an ACQUITY UPLC™ BEH C18 column (50 × 2.1 mm, 1.7 µm) equipped with Waters Quattro Premier™ and Waters ACQUITY UPLC™. The mobile phase consisted of 0.005% acetic acid in H_2_O:CH_3_CN (95:5 *v*/*v*) (A), and 0.005% acetic acid in CH_3_CN (B). The flow rate was 0.3 mL/min. A gradient mobile phase B was applied at 25% during the first 0.5 min and then increased to 100% for 7 min. Analytes were acquired using negative electrospray ionization (ESI) in multiple reaction mode (MRM). The source temperature was 120 °C, the desolvation temperature was 300 °C, the cone gas flow rate was 40 L/h, and the desolvation gas flow rate was 1000 L/h. The column temperature was set at 40 °C. The negative ionization transitions of arachidonic acid (*m*/*z* 303.3 > 259.3), 11,12-EET (*m*/*z* 319.3 > 167.3), and 14,15-EET (*m*/*z* 319.3 > 175.3) were monitored at collision energies of 15, 13, and 13 eV, respectively. QuanLynx software (ver. 4.1) was used to calculate the peak areas.

### 4.6. Docking Modeling

The docking analysis of paclitaxel into the determined structures of CYP2C8 wild-type and the L361F mutant was carried out using AutoDock 4.2 (The Scripps Research Institute, La Jolla, CA, USA), which is a grid-based docking program [28]. AutoDock uses a modified genetic algorithm that applies a local search to identify the orientations of the probe molecule and low-energy binding sites [29]. Structural coordinates of the L361F mutation were obtained from CYP2C8 (PDB ID 1PQ2) using the Coot program, and the parameters of the paclitaxel molecule were obtained from the Protein Data Bank (PDB ID 3J6G) [30]. Before docking, all water molecules were eliminated from the PDB, except for the prosthetic heme group, and only polar hydrogens were included in the model. For each docking run, a 60 × 60 × 60 grid with a spacing of 0.5 Å was used that covered the active site of the 1PQ2 structure. The results of the autodocking runs predicted a single docking location in the active site at appropriate positions for hydroxylation located near the heme iron.

## 5. Conclusions

The recombinant P450 2C8 wild-type and four variants of the enzyme were expressed in *E. coli* and their P450 2C8-mediated drug metabolism catalytic activities were analyzed in an in vitro system. As P450 2C8 is a central enzyme involved in the metabolism of many clinical drugs, as well as arachidonic acid, the differences in these polymorphisms in P450 2C8 and their impact on the metabolism of these drugs need to be precisely examined. This study suggests that individuals with these P450 2C8 allele variations are likely to have metabolic changes in clinical medicine and the production of fatty acid-derived signaling molecules. 

## Figures and Tables

**Figure 1 ijms-24-08032-f001:**
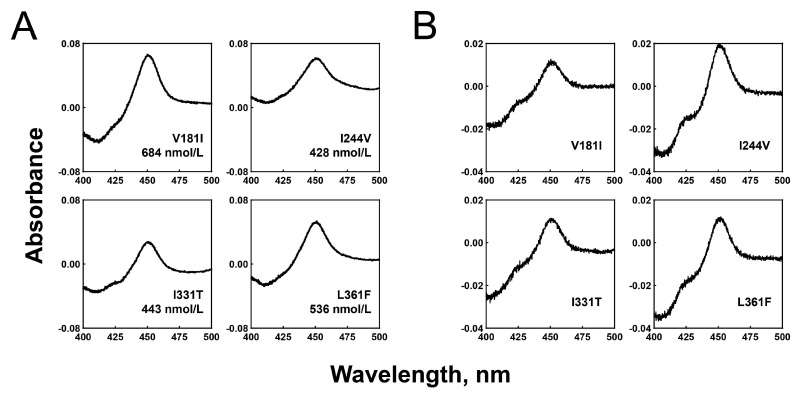
Expression and purification of recombinant P450 2C8 allelic variant enzymes in *E. coli*. (**A**) CO-binding spectra of P450 variant enzymes in *E. coli* whole cells were measured, and the contents of the P450 holoenzymes were calculated. (**B**) CO-binding spectra of purified P450 variant enzymes were measured.

**Figure 2 ijms-24-08032-f002:**
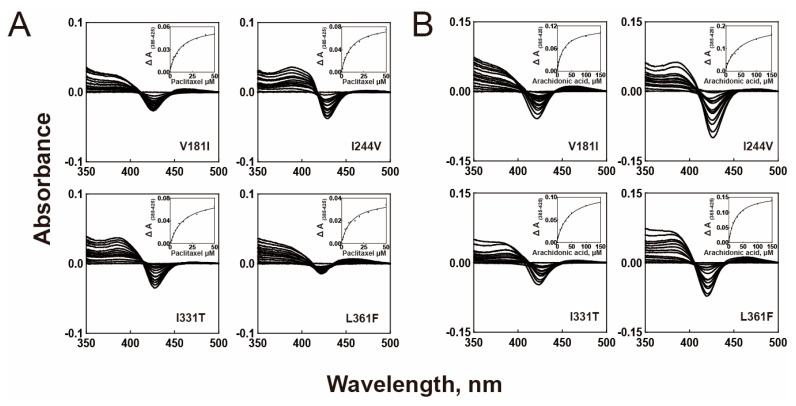
Substrate binding spectra of purified wild-type and P450 2C8 mutant enzymes. (**A**) Paclitaxel binding titration with purified P450 2C8 variants. (**B**) Arachidonic acid binding titration with purified P450 2C8 variants. The insets indicated that the difference between the maximum (385 nm) and minimum (425 nm) absorbance was plotted against the substrate concentration. The binding parameters (*K*_d_) are shown in Table 1.

**Figure 3 ijms-24-08032-f003:**
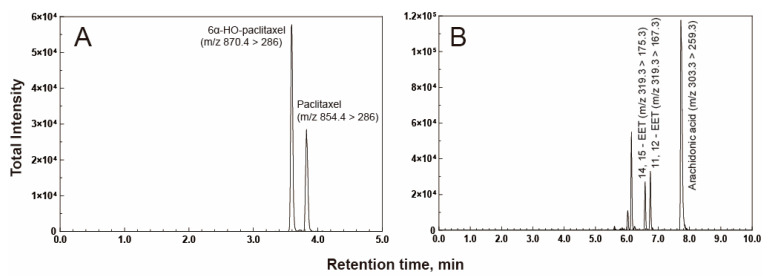
LC-MS/MS chromatogram of paclitaxel 6α-hydroxylation (**A**) and arachidonic acid epoxidation (**B**) by P450 2C8 enzymes. The analytes were observed using positive electrospray ionization and multiple reaction mode (mrm). The positive ionization transitions of substrates and products are indicated above peaks.

**Figure 4 ijms-24-08032-f004:**
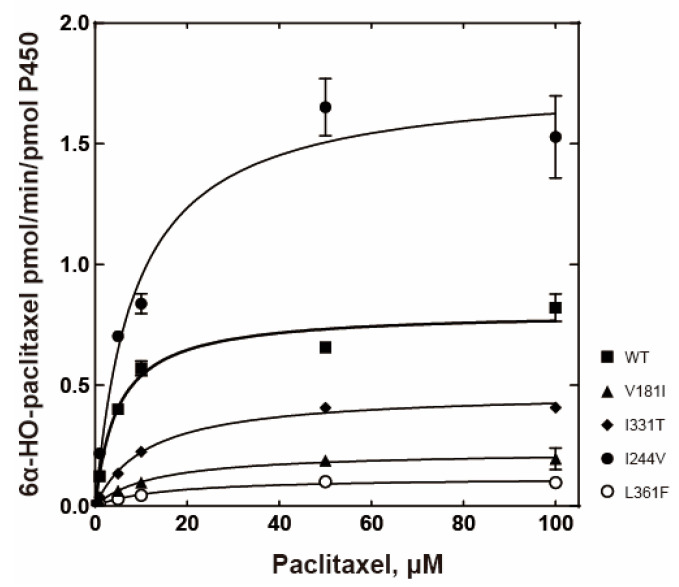
Steady-state kinetic analysis of paclitaxel 6α-hydroxylation by purified P450 2C8 variants. Steady-state kinetic parameters were obtained by fitting to a standard Michaelis–Menten equation using GraphPad Prism software (GraphPad). The steady-state kinetic parameters are shown in Table 2.

**Figure 5 ijms-24-08032-f005:**
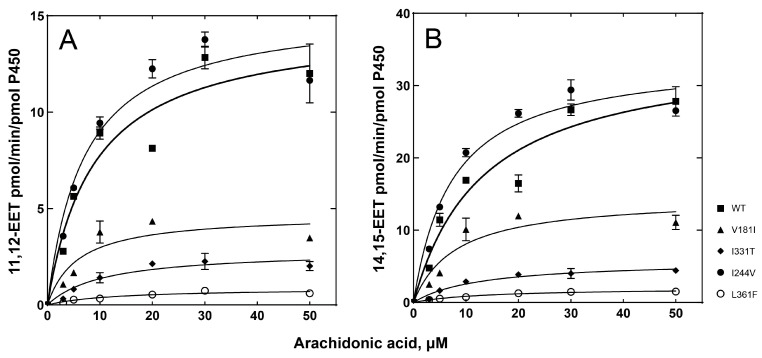
Steady-state kinetic analysis of arachidonic acid epoxidation by purified P450 2C8 variants. (**A**) Steady-state kinetic analysis of arachidonic acid ω-9 epoxidation to produce 11,12-EET. (**B**) Steady-state kinetic analysis of arachidonic acid ω-6 epoxidation to produce 14,15-EET. The steady-state kinetic parameters are shown in Table 3.

**Figure 6 ijms-24-08032-f006:**
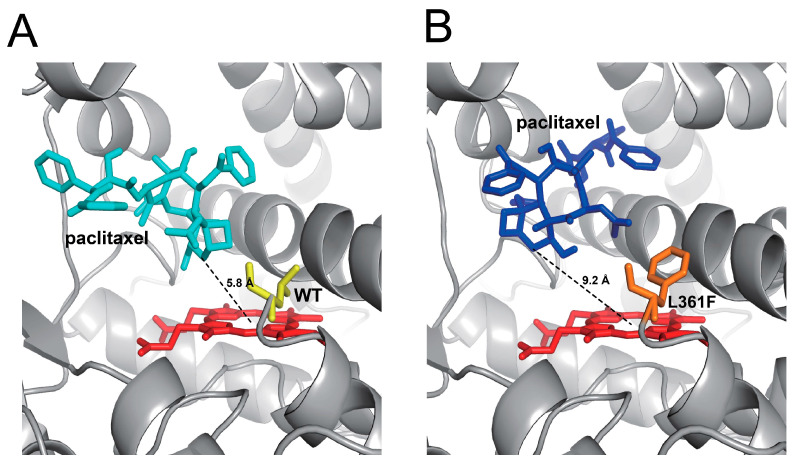
Molecular docking model of P450 2C8. The molecular docking models were constructed using the reported X-ray crystal structure of P450 2C8 (PDB ID 1PQ2). (**A**) Paclitaxel in the active site access channel of P450 2C8 wild-type. The distance between the heme of P450 2C8 and the C6 of paclitaxel was measured to be ~5.8 Å. (**B**) Paclitaxel in the active site access channel of the L361F mutation. The distance between the heme and the C6 of paclitaxel was measured to be ~9.2 Å.

**Table 1 ijms-24-08032-t001:** Substrate binding affinities of the P450 2C8 variants.

P450 2C8 Variants	Substrates	
	Paclitaxel	Arachidonic Acid
WT	10.2 ± 1.0 μM	20.3 ± 1.4 μM
V181I	10.8 ± 1.4 μM	22.5 ± 2.1 μM
I331T	13.7 ± 2.0 μM	39.1 ± 3.8 μM
I244V	10.4 ± 1.5 μM	49.0 ± 9.5 μM
L361F	10.8 ± 2.2 μM	25.2 ± 1.7 μM

**Table 2 ijms-24-08032-t002:** Catalytic activities of P450 2C8 wild-type and allelic variants for paclitaxel hydroxylation.

P450 2C8 Variants	6α-Hydroxylation of Paclitaxel			
	*k*_cat_ (min^−1^)	*K*_m_ (µM)	*k*_cat_/*K*_m_	*Relative* *Catalytic* *Efficiency*
WT	0.81 ± 0.03	4.9 ± 0.8	0.164 ± 0.028	1
V181I	0.23 ± 0.01	12.8 ± 2.7	0.018 ± 0.004	0.11
I331T	0.47 ± 0.01	11.3 ± 1.2	0.042 ± 0.005	0.26
I244V	1.77 ± 0.09	8.6 ± 1.6	0.205 ± 0.039	1.25
L361F	0.12 ± 0.01	14.8 ± 2.5	0.008 ± 0.001	0.05

**Table 3 ijms-24-08032-t003:** Catalytic activities of P450 2C8 wild-type and allelic variants for arachidonic acid epoxidation.

P450 2C8 Variants	Arachdonic Acid Epoxidation							
	ω-9 Epoxidation of Arachidonic Acid				ω-6 Epoxidation of Arachidonic Acid			
	*k*_cat_ (min^−1^)	*K*_m_ (µM)	*k*_cat_/*K*_m_	*Relative* *Catalytic* *Efficiency*	*k*_cat_ (min^−1^)	*K*_m_ (µM)	*k*_cat_/*K*_m_	*Relative* *Catalytic* *Efficiency*
WT	14.6 ± 1.3	8.8 ± 2.4	1.65 ± 0.47	1	35.1 ± 3.4	13.3 ± 3.4	2.64 ± 0.73	1
V181I	4.7 ± 0.6	5.8 ± 2.5	0.81 ± 0.37	0.49	14.6 ± 1.8	8.0 ± 2.9	1.80 ± 0.68	0.68
I331T	2.8 ± 0.3	10.7 ± 3.4	0.26 ± 0.09	0.16	5.8 ± 0.5	12.6 ± 3.1	0.46 ± 0.12	0.17
I244V	15.3 ± 1.0	7.0 ± 1.5	2.19 ± 0.47	1.33	34.0 ± 1.8	7.4 ± 1.3	4.57 ± 0.82	1.73
L361F	0.9 ± 0.1	12.9 ± 3.7	0.07 ± 0.02	0.04	2.0 ± 0.1	13.7 ± 2.2	0.15 ± 0.03	0.06

## Data Availability

All the data in this study are included in this manuscript.

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
