# Peer review of "Functional Characterization of Allelic Variations of Human Cytochrome P450 2C8 (V181I, I244V, I331T, and L361F)"

_ijms, 2023, doi:10.3390/ijms24098032_

Round 1
Reviewer 1 Report
Thank you for the authors for presenting such a great work.
Author Response
Thank you for your kind review.
Reviewer 2 Report
Generally, the paper is easy to read and is of interest to the field. The biggest thing that needs to be addressed is that the data itself does not show what the authors talk about in the results and discussion. In Figures 4 and 5, the mutant that has the highest activity is listed as I331T based on the legend and yet all of the text and tables say that it should be I244V. I suspect it is a mislabel of the filled in circle versus the diamond, but the authors should check the data to be sure!
Assuming that it is just a mislabel on all of the graphs, then the rest of the review are minor changes needed or suggested.
Figure 2- it is really hard to read the inset values- like what wavelengths are used for the difference binding. It should either be readable in the figure and/or explicitly said in the materials and methods and/or put in the figure legend. All of the places that I normally looked for that kind of information did not have it.
Lines 89-95 I suggest a rewrite of these sentences. There are several ideas that are all mixed together. I suggest changing it to:
“…displayed reduced catalytic efficiencies for paclitaxel (Fig. 4 and Table 2) In particular, the L361F variant exhibited a significant decrease in kcat and an increase in Km resulting in a catalytic efficiency of only 4.7% of that of the wild type enzyme. However, I244V exhibited a significantly a significantly enhanced catalytic efficiency, with an increased kcat value (Fig. 4 and Table 2).
Line 94- I suggest a new paragraph start with the sentence “As an endogenous substrate of P450 2C8…” since it is a separate set of data for a different substrate. I think it warrants having a new paragraph there.
Line 101 I think the figure that should be referred to is Figure 5, not figure 3, since Figure 3 is mass spec and not about catalytic efficiency directly.
Figure 4 is really BIG compared to the rest of the figures. Maybe it could be sized like the others.
Discussion
I really wanted to have some commentary on the other mutants, not just L361F and I244V. I would suggest adding a sentence or two trying to rationalize the other mutant data with reference to the type of amino acid substituted.
I would also like to see if there are any comparisons to other mutants from other species or other subfamilies that could be made. I felt like it was missing the connection to what was in the literature already. Or if nothing like it had been done before to say that outright.
Author Response
1) Generally, the paper is easy to read and is of interest to the field. The biggest thing that needs to be addressed is that the data itself does not show what the authors talk about in the results and discussion.
> We described the expected outcomes from binding data and kinetic data in Results section (pages 12, 13, and 14). Thank you.
2) In Figures 4 and 5, the mutant that has the highest activity is listed as I331T based on the legend and yet all of the text and tables say that it should be I244V. I suspect it is a mislabel of the filled in circle versus the diamond, but the authors should check the data to be sure!
> We corrected the mislabels in Fig 4 and 5. We checked the raw data and confirmed the kinetic parameters of I244V (with the highest kcat) and I331T. Thank you so much.
Assuming that it is just a mislabel on all of the graphs, then the rest of the review are minor changes needed or suggested.
Figure 2- it is really hard to read the inset values- like what wavelengths are used for the difference binding. It should either be readable in the figure and/or explicitly said in the materials and methods and/or put in the figure legend. All of the places that I normally looked for that kind of information did not have it.
> We corrected the insets in Fig. 2 more readable and described that the difference between the maximum at 385 nm and minimum at 425 nm absorbance was plotted against the substrate concentration to calculate the substrate-binding affinity (Kd) in Methods and Figure legend. Thank you.
Lines 89-95 I suggest a rewrite of these sentences. There are several ideas that are all mixed together. I suggest changing it to: “…displayed reduced catalytic efficiencies for paclitaxel (Fig. 4 and Table 2). In particular, the L361F variant exhibited a significant decrease in kcat and an increase in Km resulting in a catalytic efficiency of only 4.7% of that of the wild type enzyme. However, I244V exhibited a significantly a significantly enhanced catalytic efficiency, with an increased kcat value (Fig. 4 and Table 2).
> We corrected these sentences as a reviewer suggested. Thank you so much.
Line 94- I suggest a new paragraph start with the sentence “As an endogenous substrate of P450 2C8…” since it is a separate set of data for a different substrate. I think it warrants having a new paragraph there.
> The description on the kinetics of arachidonic acid started in the separate paragraph as a reviewer suggested. Thank you.
Line 101 I think the figure that should be referred to is Figure 5, not figure 3, since Figure 3 is mass spec and not about catalytic efficiency directly.
> We corrected these sentences as a reviewer suggested. Thank you so much.
Figure 4 is really BIG compared to the rest of the figures. Maybe it could be sized like the others.
> We made a new Figure 4 with appropriate sizes. Thank you so much.
Discussion
I really wanted to have some commentary on the other mutants, not just L361F and I244V. I would suggest adding a sentence or two trying to rationalize the other mutant data with reference to the type of amino acid substituted.
> Our postulation of these two mutants were proposed from the structure of P450 2C8. The mutation of V181I was located at the C-terminus of helix E of P450 2C8 structure {Schoch, 2004 JBC PMID 14676196}. The Val181 residues in the helix E appears to play a role in interacting with helix G and the N-terminus of helix H. Therefore, the additional methyl chain of V181I mutant could produce the extra space between helices and perturbate the optimal orientation of helices E, G, and H albeit substitution of similar hydrophobic residue. Meanwhile, the I331T mutation was found at the C-terminus of J-helix {Schoch, 2004 JBC PMID 14676196}. The Ile331 reside is close to the Arg333, which may be one of the basic residues interacting with the NADPH-P450 reductase {Lee, 2023 JIB PMID 36640554}. Therefore, I331T mutation could affect the reductase binding and electron transfer step, which should be further analyzed.
We describe this in the Discussion section with additional supplementary figure (pages 15-16).
I would also like to see if there are any comparisons to other mutants from other species or other subfamilies that could be made. I felt like it was missing the connection to what was in the literature already. Or if nothing like it had been done before to say that outright.
> Professor Halpert reported the role L362F mutant of CYP2B1 (corresponding to L361F of CYP2C8) in oxidation of steroids and 7-alkoxylcoumarins (Domanski, et al 2001 Arch Biochem Biophys 394, 21). The study showed the dramatic decrease of enzyme activity in L362F mutant of CYP2B1. We discussed this comparison in Discussion section (page 15). Thank you.

Reviewer 3 Report
In this paper, the authors studied the substrate binding affinities and the catalytic activities of the cytochrome P450 2C8 wild type and its allelic variants. From this study they postulate the influence of these mutants to the drug metabolism and production of fatty acid-derived signal molecules.
Please address the following points:
1. In figure 1, the authors claim they don't observe any peak at 420nm but the figure 1B clearly shows there is a peak buried in 420nm. Can you please clarify. Please include the wild type spectra to this panel to confirm the structural stability is same between the wild type and its mutants. Also please explain why the peak maximum at 450 nm is very different between the mutants. Please use the same scale of absorbance in all the panels to distinguish the differences.
2. Km and Kcat/Km, the kinetic parameters obtained from fitting to a Michaelis–Menten equation shows very high error (>15-47%). So the relative catalytic efficiency calculated using the current parameters is highly influenced by errors so they are not accurate. This needs to be addressed by including more data points (followed by fitting) for both the steady-state analysis of paclitaxel 6α-hydroxylation and arachidonic acid epoxidation.
Round 2
Reviewer 2 Report
I think that the changes that were made address all of my previous concerns.